# An Outbreak of Newcastle Disease Virus in the Moscow Region in the Summer of 2022

**DOI:** 10.3390/vetsci10060404

**Published:** 2023-06-19

**Authors:** Artyom Rtishchev, Anastasia Treshchalina, Elena Shustova, Elizaveta Boravleva, Alexandra Gambaryan

**Affiliations:** 1Federal State Budgetary Scientific Institution «I. Mechnikov Research Institute of Vaccines and Sera», 105064 Moscow, Russia; 2Chumakov Federal Scientific Center for the Research and Development of Immune-and-Biological Products, Village of Institute of Poliomyelitis, Settlement “Moskovskiy”, 108819 Moscow, Russiashustva_eu@chumakovs.su (E.S.);

**Keywords:** NDV, Paramyxovirus, outbreak, pathogenicity

## Abstract

**Simple Summary:**

An outbreak of Newcastle disease viruses AAvV-1 subgenotype VII.1 was described in the Moscow region of Russia in 2022. The outbreak occurred on a private backyard farm located far from other poultry farms. The virus was extremely pathogenic and contagious in chicken, while it was virtually harmless to mice. The epidemic situation of Paramyxovirus in Russia is discussed.

**Abstract:**

In August 2022 on a backyard farm in the Moscow region of Russia, mortality was observed among chickens, and all 45 birds of a particular farm died or were slaughtered after the onset of symptoms within a few days. Paramyxovirus was isolated from the diseased birds. Based on the nucleotide sequences of the F and NP gene fragments, it was determined that the virus belonged to subgenotype VII.1 AAvV-1 class II. The cleavage site of the F gene _109_SGGRRQKRFIG_119_ and T in 546 and 555 position of the NP gene were typical for the velogenic type. The genetically closest NDV isolates were found in Iran. The mean time of death of 10-day-old chicken embryos upon infection with the minimal infectious dose was 52 h, which is typical for the velogenic pathotype. The virus caused 100% death of six-week-old chickens during oral infection as well as 100% mortality of all contact chickens, including those located in remote cages, which proves the ability of the virus to spread not only by the fecal–oral route but also by the aerosol route. That demonstrates a high level of pathogenicity and contagiousness of the isolated strain for chicken. However, mice intranasally infected with high doses of the virus did not die.

## 1. Introduction

Paramyxoviruses (family *Paramyxoviridae)* have a wide host range, including mammals and birds. Avian viruses of the subfamily *Avulavirinae* (AAvV) consist of 22 serotypes [1]. More than 200 species of wild birds belonging to 27 orders are the natural reservoir of *Avulavirinae*. Seasonal migrations of wild birds contribute to the widespread distribution of the pathogen, and transmission and spread of the virus is possible both within the same species and between species. The best known among them are viruses of the first serotype, the so-called Newcastle disease viruses (NDVs), which have caused panzootics five times, the last of which is still ongoing. Newcastle disease is a highly contagious disease that causes incalculable economic losses and poses a serious threat to the poultry industry worldwide. The uncontrolled live bird trade and live bird markets are important sources for the spread of dangerous viruses, including both NDV and highly pathogenic avian influenza viruses [2].

The current classification of NDV is based on the F fusion protein gene. NDVs are divided into two classes (class I and class II). Class I includes generally low-virulence strains isolated from wild waterfowl, and class II includes viruses of both low and high virulence isolated from domestic and wild birds.

The genome of Paramyxoviruses is single-stranded RNA with negative polarity, meaning it is complementary to the mRNA encoding viral proteins. The genome encodes the viral products: nucleoprotein (NP), phosphoprotein (P), matrix (M), fusion (F), hemagglutinin-neuraminidase (HN), and RNA polymerase (L). During transcription of the P gene, two more proteins are synthesized—the V protein and the W protein. Antibodies to surface proteins HN and F are neutralizing and represent the primary protective component induced by Newcastle disease (ND) vaccines.

By pathogenicity for chickens, NDVs are divided into apathogenic (avirulent), weakly virulent (lentogenic), medium virulent (mesogenic), and highly virulent (velogenic). Velogenic strains are subdivided into viscerotropic and neuroptropic pathotypes based on tissue tropism. Low-pathogenic strains usually replicate in a limited range of tissues, such as the respiratory tract and intestines. Velogenic NDVs in contrast, have acquired the ability to spread throughout the body and cause respiratory diseases with diarrhoea and nervous manifestations. In poultry, the virus is transmitted primarily through the respiratory route.

The pathogenicity of the virus is largely determined by the structure of the cleavage site of the fusion protein. Initially, the F protein is synthesized as an inactive F0 protein, which is subsequently cleaved into F1 and F2 subunits by host proteases. Low-pathogenic viruses have a _109_SGGGR(K)QGRLIG_119_ cleavage site. The F protein of such viruses can only be cleaved by extracellular trypsin-like proteases of the respiratory and gastrointestinal tracts. In highly pathogenic viruses, the site of proteolysis has several basic amino acids (lysine or arginine) and phenylalanine at amino acid position 117 of _109_SGGRRQ(K/R)RF(V/I)G_119_. Cleavage of the F0 protein at this site is possible by furin-like proteases present in all cells of the body. Such viruses cause a generalized infection. Pathological manifestations of genotype VII class 2 NDV infection in birds include multi-organ hemorrhage, neurological symptoms, and death [3].

Non-pathogenic AAvVs have been described by many researchers who have studied avian influenza viruses because they occur in the same hosts and are isolated in a similar manner. Pathogenic NDVs have been isolated from poultry and pigeons. Outbreaks of NDVs mostly occur in developing countries, affecting both commercial and backyard flocks.

Figure 1 shows the cases of AAvV isolation in Eurasia and Africa described in recent years in the literature referred by Pubmed accessed on 22 November 2022 (https://pubmed.ncbi.nlm.nih.gov). The serotypes, classes, and genotypes as well as the year of isolation and the source of isolation are indicated: wild birds (W), pigeons (Pi), and poultry (Ch). If the pathogenicity of the virus was established in the cited work, then it is indicated by color: red indicates pathogenic isolates and green indicates non-pathogenic isolates [4,5,6,7,8,9,10,11,12,13,14,15,16,17,18,19,20,21,22,23,24,25,26,27,28,29,30,31,32,33,34,35,36,37,38,39,40,41,42,43,44,45,46,47,48,49,50,51,52,53,54,55,56,57,58,59,60,61,62,63,64,65,66,67,68,69,70,71,72].

AAvVs are widely distributed around the world. With the exception of exotic serotypes 17, 18, and 19 isolated from Antarctic penguins, related viruses have been found on all continents, both in the western and eastern hemispheres.

A total of 57 APMV isolates belonging to species AAvV-1, AAvV-2, AAvV-4, AAv -6, AAvV-12, AAvV-21, and AAvV-22 were isolated from wild birds and domestic poultry during 2009–2020 in Taiwan. Some Taiwanese wild bird AAvV -1 and AAvV -4 viruses have been introduced from North America. An NDV of sub-genotype VI.2.1.1.2.2 was isolated from pigeons, and an NDV of sub-genotype VII.1.1 was isolated from domestic chicken. The F cleavage sites of these viruses were _112_(K/R)RQKR↓F_117_ and _112_RRKKR↓F_117_, respectively. The chicken virus was related to viruses that caused the Newcastle disease epidemic in Taiwan. The pathogenicity status of these viruses was not reported in this study [25].

Over the past 10 years, numerous cases of isolation of pathogenic NDVs have been described in African countries, Iran, Pakistan, Kazakhstan, India, China, Bangladesh, and Malaysia. Subgenotypes VI, VII, and XIII dominated among pathogenic NDV, and subgenotypes XXI and XVIII also occurred.

At the same time, in Europe and Japan only apathogenic AAvVs 1, 4, 6, 7, 12, 14, 21, and 22 serotypes have been detected. Apathogenic AAvVs 1, 4, and 6 serotypes have been described in Russia. Also in Russia, pathogenic NDV XXI, VI, and 1.II VII genotypes were isolated from pigeons, but the last of these cases dates back to 2011.

The lack of registration of pathogenic NDVs in the developed countries of Europe and Asia is possibly due to industrial poultry farming and a well-established system of control and vaccination. However, in Russia, there are many small backyard flocks where vaccination of poultry is not carried out. This makes possible the penetration of pathogenic NDVs into Russia. In this paper, we describe a case of a local outbreak of NDV in the Moscow region in 2022.

## 2. Materials and Methods

### 2.1. Reagents

MycoKill AB was from PAA Laboratories GmbH, Pasching, Austria. Viral RNA Mini Kit was from QIAGEN, Hilden, Germany. MMLV Reverse Transcription kit, nuclease-free water, random primers, TAE buffer, and DNA Ladder were from Evrogen, Moscow, Russia. Ribonuclease inhibitor was from Syntol, Moscow, Russia. Sequencing Reagent Kit ABI PRISM^®^ BigDye™ Terminator v. 3.1 were from ThermoFisher Scientific, Waltham, MA, USA.

### 2.2. Animals

Ten-day-old embryonated chicken eggs (CE) were purchased from the Ptichnoye State Poultry Farm, Moscow, Russia. Leghorn chickens aged 40–45 days without antibodies to NDV, obtained from a poultry farm free from infectious avian diseases were used. (Poultry farm Tomilinskaya, Moscow, Russia). Two-month-old BALB/c mice were from the Lesnoye farm, Moscow, Russia. All tests were carried out in compliance with the standard for keeping and care of laboratory animals GOST 33215-2014, adopted by the Interstate Council for Standardization, Metrology and Certification as well as in accordance with the requirements of Directive 2010/63/EU of the European Parliament and of the Council of the European Union of 22 September 2010 on the protection of animals used for scientific purposes. All experiments involving work with live viruses were carried out in a biosafety level 3 facility.

### 2.3. Isolation of the Virus

Lung and kidney samples from diseased and slaughtered chickens were used to isolate the virus. A solution of kanamycin (0. 01 mg/mL), nystatin (0.1 mg/mL), gentamicin (0.4 mg/mL), and 2% MycoKill AB in phosphate-buffered saline (PBS) was prepared. Then, 0.5 g of tissue samples were triturated with fine glass, and 2 mL of antibiotics solution was added. The suspensions were centrifuged, and the supernatants were inoculated into CE. The eggs were examined twice a day and transferred to the refrigerator after the death of the embryo. A hemagglutination assay with 1% chicken erythrocytes was carried out with all collected allantoic fluid, and the amount of virus was expressed in hemagglutination units. The infected allantoic fluid (IAF) was aliquoted, frozen, and used in subsequent work. The 50% infectious dose (EID_50_) in the samples was determined by CE titration. Protocols for virus isolation and titration followed OIE standards [73].

### 2.4. Sequencing and Phylogenetic Analysis

The viral RNA was extracted from IAF using QIAamp Viral RNA Mini Kit (QIAGEN, Hilden, Germany) following the manufacturer’s instructions. The reverse transcription reaction was carried out using the MMLV RT kit (Evrogen, Moscow, Russia) in the presence of a random decanucleotide primer. Polymerase chain reaction was carried out with the Tersus Plus PCR Kit (Evrogen, Moscow, Russia) in a volume of 25 µL, where the following components were used: sterile water for PCR—17.5 µL; 10× Tersus Plus buffer—2.5 µL; 50× dNTP mix—0.5 µL; forward primer fFapmv2 (10 μM)—1 μL; reverse primer rFapmv2 (10 µM)—1 µL, cDNA—2 µL; and 50× Tersus polymerase—0.5 µL. Oligonucleotides used in the work: fFapmv2-(ATGGGCTCCAGACCTTCTAC); rFapmv2-(CTGCCACTGCTAGTTGCGATAATCC); fNPapmv-(GGTATTCTGTCTTCGGATTG); and rNPapmv-(TCATCCGATATAAACGCAT). The nucleotide sequence of the gene fragments were obtained by Sanger sequencing on an ABI PRISM 3130 Genetic Analyzer (Applied Biosystems, ThermoFisher Scientific, Waltham, MA, USA) using an ABI PRISM^®^ BigDye™ Terminator v. 3.1 (ThermoFisher Scientific, Waltham, MA, USA). Analysis of the PCR results was performed by electrophoresis in a 2% agarose gel in Tris-acetate buffer. PCR fragments about 500 bp were excised for gel purification with the Qiagen MinElute Gel Extraction Kit (QIAGEN, Hilden, Germany) according to the manufacturer’s instructions. The complete F gene coding nucleotide sequences of class II NDV isolates were downloaded from GenBank of the National Center for Biotechnology Information accessed on 3 October 2022 (https://www.ncbi.nlm.nih.gov/genbank/). Sequence processing was performed using BioEdit 7.2. and MEGA X (https://bioedit.software.informer.com/, accessed on 3 October 2022) and (https://www.megasoftware.net/, accessed on 3 October 2022). All nucleotide sequences were aligned using the MUSCULE algorithm and cut in the reading frame. Maximum-likelihood trees based on a general time-reversible (GTR) model were constructed by using the BEAST software package (1.10.4). Analysis was run over 10,000,000 generations, and trees were sampled every 1000 generations, resulting in 10,000 trees. The iTOL v6 online service (https://itol.embl.de/, accessed on 3 October 2022) was used to visualize and annotate the tree. Genotype identification was carried out on the basis of phylogenetic topology.

### 2.5. Mean Death Time (MDT) Assays

A 10-fold serial dilution of fresh IAF in sterile PBS was performed. Then, 0.1 mL of each dilution was inoculated into five 10-day-old chicken embryos. The eggs were incubated at 36 °C and were observed three times a day for 5 days. The times of any embryo death were recorded. The highest virus dilution that caused 100% mortality was considered as the minimum lethal dose. The MDT was the mean time for the minimum lethal dose to kill all the inoculated embryos.

### 2.6. Analysis of the Pathogenicity and Contagiousness of the Virus in Chicken

Three groups of five six-week-old chickens were formed. Then, 10^8^ EID_50_ of the NDV6081virus was added into the drinker of the first group. The next day, the chickens of the first group were placed together with the chickens of the second group. The chickens of the third group were kept in a cage located two meters from cages with infected birds.

### 2.7. Analysis of the Pathogenicity of the Virus in Mice

Groups of six BALB/c mice were anesthetized and inoculated intranasally with placebo or diluted IAF in a volume of 50 µL with doses of 10^3^, 10^4^, 10^5^, and 10^6^ EID_50_ per mouse. Survival and body weight of mice were monitored daily.

### 2.8. Ethics Statement

A total of 30 chickens and 60 mice were used in the study. All experiments were performed in accordance with the (European Convention for the Protection of Vertebrate Animals used for Experimental and Other Scientific Purposes, Strasbourg, 18 March 1986). All necessary measures were taken to alleviate the suffering of animals. The study design was approved by the Ethics Committee of the Chumakov Federal scientific center for the research and development of immune-and-biological products, Moscow, Russia. (Approval #4 from 2 December 2014)

## 3. Results

In August 2022, on a backyard farm in the Moscow region of Russia, the Chernogolovka district, loss of chickens suddenly began. The poultry yard was located two kilometers from the nearest poultry farms. Chickens were not vaccinated against NDV. All 45 birds of this farm became diseased and died within a few days. Clinical signs of diseased chickens were the discharge of gray mucus from the nostrils and beak, sharp coughing sounds, depression, twisted head, and diarrhea. The chickens died 1–3 days after the onset of symptoms. At autopsy, extensive hemorrhages in the lungs and sharply enlarged kidneys were found. Lung and kidney tissue samples were taken from slaughtered birds for virus isolation. All chicken embryos inoculated with tissue samples died within 30 h. All IAFs were positive in the hemagglutination assay. The NDV virus was detected in all material samples by RT-PCR. The virus was named as NDV/Chicken/Moscow/6081/2022 (NDV6081). Fragments of fusion and NP genes were sequenced and used for genotyping. GenBank accession numbers are OQ190211 and OQ190212. The cleavage site of NDV6081 was _109_SGGRRQKRFIG_119_, which is characteristic of pathogenic viruses.

Additionally, the nucleotide sequence of the NP gene fragment encoding the nucleocapsid protein (697 nt) was obtained. Positions 546 and 555 of the NDV6081 NP gene correspond to the velogenic variant (T at position 546 and T at position 555) [74].

### 3.1. Phylogenetic Analysis of the NDV/Chicken/Moscow/6081/2022

To construct an evolutionary tree, samples of the F gene sequences of each class 2 genotype (*n* = 125) and fragments of the F gene of the Moscow isolate and 24 nearest viruses identified by BLAST were used. Analysis of the phylogenetic relationships of the nucleotide sequences encoding the F gene of class II NDV isolates showed that the studied NDV6081 isolate belongs to the AAvV-1 genotype VII. The closest relatives are chicken viruses isolated in Iran in 2011–2013 (97.03–97.48% similarity) (numbers from GenBank KU201408-10, KU201413-15 and MZ463065), these viruses were included in one cluster with a posterior node probability of 0.98 (Figure 2). Additionally, these viruses were included in a larger cluster consisting of viruses isolated from Iran in 2012–2015. Since the date of isolation of viruses from Iran dates back 10 years before the isolation of NDV6081, it cannot be concluded whether the virus was brought into Russia directly from Iran or through an intermediate location.

### 3.2. Pathogenicity and Contagiousness of the NDV/Moscow/6081/2022 for Chickens

To determine the median time to death (MDT) of NDV6081, sets of 10-day-old chicken embryos were infected with infectious allantoic fluid at dilutions of 10^−4^, 10^−5^, 10^−6^, and 10^−7^. The eggs were observed three times a day, and the time of death was recorded. In the last set, some of the embryos survived. The median survival time in the set challenged with IAF at a dilution of 10^−6^, which was taken as the minimum infective dose, was 52 h. MDT 52 h corresponds to the velogenic type (up to 60 h).

To assess the pathogenicity of the NDV6081 for six-week-old chickens, three groups of five birds were formed: infected, direct contact, and remote contact. Chickens were housed in three cages in a biosafety level 3 facility. On day zero 10^8^ EID_50_ of the NDV6081virus was added to the drinker of the first group (infected). After 5 h, the drinker was removed. The next day, the chickens of the infected group were transferred to the cage with the chickens of the direct contact group, and the first cage was removed for disinfection. The chickens of the remote contact group were kept in a distant cage located two meters from cages 1 and 2. The contamination by feed particles and feces were excluded, yet the airflow and the transfer of fine dust between the cages were possible.

The dynamics of the death of infected and contact chickens is shown in Table 1.

All infected chickens died by the fifth day, all chickens from direct contact group died on the sixth day, and the chickens of the remote contact group became diseased and began to die on day seven, after which they all died by the tenth day. The organs and feces of the chickens of remote contact group were examined for the presence of the virus. In all tested samples (in the brain, lungs, kidneys, intestines, and feces) the NDV was detected by PCR.

That is, with direct contact (keeping in one cage), chickens became infected almost immediately, while the death of chickens in a distant cage was probably determined by the infection of one of the chickens through the air flow, which initiated infection and rapid death of the whole group. The result of this experiment demonstrates the very high pathogenicity and contagiousness of NDV6081. Infection occurred not only by the fecal–oral route but also through fine dust.

### 3.3. Pathogenicity of the NDV/Moscow/6081/2022 for Mice

Groups of mice were infected with 10^3^, 10^4^, 10^5^, and 10^6^ EID_50_ per mouse, respectively. The control group received a placebo. Survival and weight of mice were recorded within 12 days after infection. No mice died during the experiment (Table 2). In groups infected with high doses of the virus, there was a slight lag in weight on the second to fifth day after infection. However, by day 12, all mice were practically healthy. Antibodies to NDV were found in the sera of infected mice. Thus, the NDV6081 virus was practically not pathogenic for mice, despite the very high pathogenicity for chickens.

## 4. Discussion

*Avulavirus* is widespread throughout the world, mainly causing asymptomatic infections in wild birds. However, *Avulavirus* of the first serotype, the so-called Newcastle disease virus, often become poultry viruses. Newcastle disease is a highly pathogenic and contagious disease that causes enormous losses to the poultry industry.

Thus, the NDV6081 virus was highly pathogenic for chickens. At the same time, it was practically not pathogenic for mice. This distinguishes NDV from highly pathogenic avian influenza viruses (HPAIV). In the latter, high pathogenicity for chickens usually correlates with increased pathogenicity for mammals. The HPAIVs are usually fatal in lab mice, ferrets, and farmed minks. They have also caused deadly disease outbreaks in tigers, marine mammals, and wild red foxes [75]. Outbreaks of H5N1 and H7 viruses were often accompanied by human illnesses, which were sometimes fatal.

On the other hand, there have been no reports of mammalian pathogenicity of avian Paramyxoviruses. This raises questions about the different mechanisms of host range and pathogenesis in NDV and HPAIV. In recent years, pathogenic strains of NDV genotype VII have been widely circulating in Africa and Asia [3]. There have been no registered cases of NDV of this subgenotype in industrial poultry farming in Russia up to 2018. In most countries, including the Russian Federation, vaccination of commercial poultry against NDV is mandatory [76]. According to the notifications of the Veterinary Service of the Russian Federation in industrial poultry farms of Russia, the Newcastle disease is classified as a controlled infection. However, NDV registration data comes from a limited number of regions in Russia, which creates a distorted picture of virus circulation in the country as a whole. The isolation of highly pathogenic NDV in the Moscow region raises the question of the circulation of these viruses in Russia. Was this an isolated case? How could the virus get to a remote backyard from other poultry farms? We do not have information on outbreaks of NDV in this region, but rumors had circulated among the residents about the death of chickens in nearby villages. We assume that the virus was introduced by wild birds, most likely jays and magpies, which fed freely along with chickens.

Although the literature cited in Pubmed does not contain data on the isolation of pathogenic NDV in Russia in recent years, information on this topic can be found in local journals. Despite regular preventive vaccination programs, the registration of new cases of the disease in poultry farms and among poultry occurs annually. In 2019, in the Russian Federation, there was a sharp increase in cases of Newcastle disease in birds with the spread of subgenotype VII-L virus throughout the country—from Primorsky Krai to Kursk Region. Many outbreaks were registered in disadvantaged areas (18 affected territories) where unvaccinated livestock were kept in the backyards of citizens. A total of 15 outbreaks of NDV were registered in 2022, and 3 outbreaks have already been registered in 2023. Comparison of the results of monitoring ND seroprevalence in wild birds in 2017 and 2019 showed a sharp increase in the proportion of immune birds. In 2017, antibodies to NDV were found in about 5% of birds tested, and they were found in 70–100% of birds in 2019, depending on the bird species. Also in 2019, serological monitoring revealed a high percentage of positive samples among domestic ducks that were not vaccinated against NDV [77,78].

ND still constitutes a potential threat to Russian poultry farming. The epizootic situation for Newcastle disease in Russia has been exacerbated in recent years, and there is a risk of repeated outbreaks, especially in the absence of routine vaccination. The aggravation of the epidemic situation in 2019 and the spread of genotype VII NDV throughout the Russian Federation raises the question of improving the practice of vaccinating chickens both in industrial poultry farming and in private backyards [76].

## 5. Conclusions

An outbreak of highly pathogenic AAvV-1 (Newcastle disease virus) was described in the Moscow region of Russia, in the summer of 2022. The outbreak was observed in a single backyard, located remote from other poultry farms. It is likely that the virus was introduced by wild birds. The virus was extremely pathogenic and contagious in chickens, while it was virtually harmless to mice.

## Figures and Tables

**Figure 1 vetsci-10-00404-f001:**
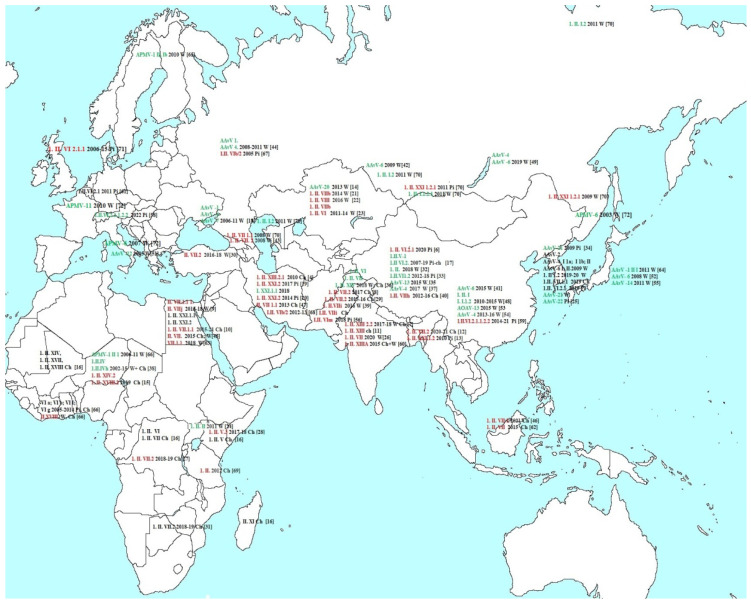
The cases of AAvV isolation in Eurasia and Africa. The serotypes, classes, and genotypes are indicated: red for pathogenic isolates and green for non-pathogenic isolates. Source of isolation: wild birds (W), pigeons (Pi), and poultry (Ch). If one work provides data on several viruses, then the link to the work is affixed only for the last isolate.

**Figure 2 vetsci-10-00404-f002:**
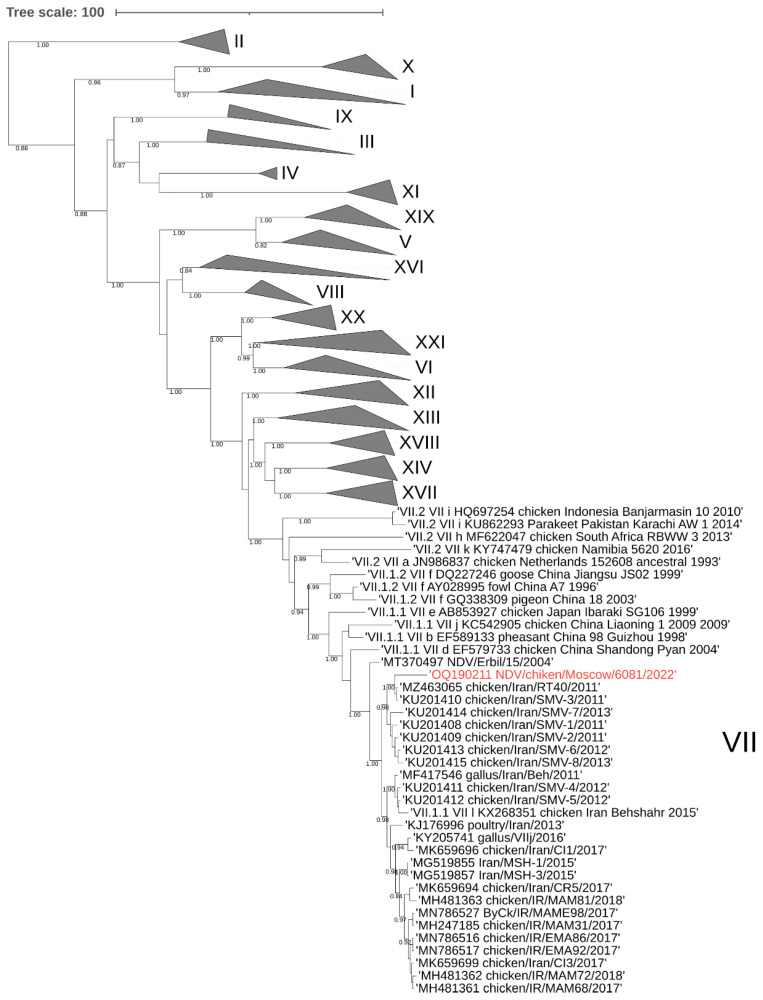
Phylogenetic tree of the genome region encoding the AAvV-1 class II fusion protein. The studied strain (isolate NDV/chicken/Moscow/6081/2022) is marked in red. Tree nodes with posterior probability >0.75 are marked in the figure. Viruses of genotypes not associated with the study are grouped.

**Table 1 vetsci-10-00404-t001:** Dynamics of death of infected and contact chickens.

Group	Mortality *	Mdd **
Infected	5/5	5
Direct contact	5/5	6
Remote contact	5/5	8.6

* Number of dead/initial number of chickens. ** Mean day of death relative to day zero.

**Table 2 vetsci-10-00404-t002:** Survival and weight dynamic of mice infected with the NDV/Moscow/6081/2022.

	Infection Dose: EID_50_ Per Mouse				
	0	10^3^	10^4^	10^5^	10^6^
Mortality *	0/6	0/6	0/6	0/6	0/6
Weight **	100%	100%	97%	96%	94%

* Number of dead/initial number of mice. ** Mean weight on day 3 relative to control group.

## Data Availability

The sequences from the study are available in GenBank (accession numbers OQ190211 and OQ190212). The data that support the findings are available from the corresponding author upon request.

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
