# Peer review of "An Outbreak of Newcastle Disease Virus in the Moscow Region in the Summer of 2022"

_vetsci, 2023, doi:10.3390/vetsci10060404_

Round 1

Reviewer 1 Report

This is a very well conceived, high-quality research report, that provides sufficient detail and data to support the findings.  The conclusions follow the data and analyses presented here. Well done.

There are some slight errors word choice and grammar throughout, eg. [abstract] "...the mice infected with high doses of the virus had not die." [line 77] "Pathogenic NDVs were been isolated..." [line 280] "In all tested sumples ..." etc.  All of these can easily be found and corrected by an experienced copy editor.

Author Response

We are grateful to the reviewer for the high appreciation of our work. We additionally checked the text of the manuscript with the help of a native English speaker.

Reviewer 2 Report

vetsci-2394472-peer-review-v1

Title:An outbreak of Newcastle disease virus in the Moscow region in the summer of 2022

This paper is a report of an investigation of an outbreak of NDV in a backyard farm in the Moscow region of Russia. It is a case report in a limited area; the background of the farms affected by NDV, including vaccine history, is unclear, and the factor basis for the spread of NDV should be clarified. The following are comments throughout.

Comments:

The introduction should be focused and concise in its description.

The background of the poultry farms inspected in this study is unclear. It should be specifically mentioned in the Materials and Methods section.

If there are vaccinated chickens, the percentage of infection and mortality rate of the test virulent strain NDV6081 to those chickens should be clearly stated.

A discussion is included in the description of the results. The discussion based on the results should be moved.

Regarding the origin of the virus:

Since the virus was similar to the Iranian isolate, the possibility of chicken distribution and migratory birds flying to and from Iran should be discussed. Jay birds are mentioned as a factor, but no clear rationale is provided.

Regarding vaccines:

The authors point out that many vaccines, such as La Sota and B1, which are widely used in many countries, have been shown to not always be highly efficient against modern strains of NDV, but the current outbreak cannot derive from the results any evidence that existing vaccines are ineffective, since no vaccine history is mentioned.

Furthermore, in their discussion, the authors state that the outbreak is controlled in developed countries in Europe and Asia because quarantine and vaccination systems have been established, which contradicts the above expression.

On the other hand, there are many poultry farms in Russia that do not have adequate poultry vaccination systems. Although the paper is an outbreak report, are the NDV cases identified in this study farms that were able to respond to the vaccine? It will be necessary to discuss the outbreak and the vaccine, incorporating background factors and suggestions for specific countermeasures.

Author Response

We are grateful to the reviewer for useful comments that will help us improve the manuscript.

We added information about the vaccine status of chickens in study farm to the text

We have clarified the background of the inspected poultry farm.

The discussion based on the results was moved in discussion section.

Regarding vaccines: - we agree with the reviewer that our work does not provide any information about the effectiveness of vaccines, so we removed the relevant section from the discussion.

Round 2

Reviewer 2 Report

It is confirmed that some revisions were made in response to comments on the initial draft, but the authors cannot confirm the revised draft in response to the comments because there are no specific sections explaining the details of the revisions. In addition, there are many areas that need to be corrected, such as the paragraph structure of M&M and the presentation of tables.

Author Response

In  point-to-point response to Reviewer 2, I have given all the changes in the text that were made on the advice of the reviewer.
In the second review by reviewer 2, there were no additional specific comments on the text
The paragraph structure of M&M and the presentation of tables are also corrected in the final manuscript.
All the advice of the reviewers was taken in the preparation of the final version of the manuscript.
